# Change-Point Detection in a High-Dimensional Multinomial Sequence Based on Mutual Information

**DOI:** 10.3390/e25020355

**Published:** 2023-02-14

**Authors:** Xinrong Xiang, Baisuo Jin, Yuehua Wu

**Affiliations:** 1School of Management, University of Science and Technology of China, Heifei 230026, China; 2Department of Mathematics and Statistics, York University, Toronto, ON M3J 1P3, Canada

**Keywords:** change-point, mutual information, likelihood ratio, high-dimensional multinomial sequence

## Abstract

Time-series data often have an abrupt structure change at an unknown location. This paper proposes a new statistic to test the existence of a change-point in a multinomial sequence, where the number of categories is comparable with the sample size as it tends to infinity. To construct this statistic, the pre-classification is implemented first; then, it is given based on the mutual information between the data and the locations from the pre-classification. Note that this statistic can also be used to estimate the position of the change-point. Under certain conditions, the proposed statistic is asymptotically normally distributed under the null hypothesis and consistent under the alternative hypothesis. Simulation results show the high power of the test based on the proposed statistic and the high accuracy of the estimate. The proposed method is also illustrated with a real example of physical examination data.

## 1. Introduction

The change-point problem was first proposed by Page [1]. It considers a model in which the distribution of the observed data changes abruptly at some point in time, which is common in biology [2], finance [3], literature [4] and epidemiology [5]. Change-point detection can be employed as a tool in time series segmentation. A typical reference in the field of study is [6]. Once a change-point is detected in a data sequence, it is used to split the data sequence into two segments so that both segments are modeled separately. On the other hand, from a practical point of view, behavior and policies can be adjusted based on changes in events of interest. So, it is very important to perform change-point detection.

There are mainly two problems in the change-point model: checking the existence of change points and estimating the positions of these change points. These issues have been studied in substantial literature. For example, see Sen and Srivastava [7] for the mean change in a normal distribution and Worsley [8] for a change in an exponential family using the maximum likelihood ratio method. Others include Bai [9] for the least squares estimate of mean shift in linear processes, Vexler [10] for the change-point problem in a linear regression model and Gombay [11] for the change-point in autoregressive time series, etc. See [12,13,14] for details.

A study has shown that most work on change-point problems has been done for continuous data [14]. In real life, however, many data are observed on a discrete scale. Common discrete distributions include binomial, multinomial and Poisson distributions. In this article, we consider the change-point problem in a multinomial sequence, which originated from the transcription of the Gospels [15]. The Lindisfarne Gospels were divided into several sections, assuming that only one author contributed to the writing of any section and that the sections written by any one author were continuous. The goal was to test whether a single author wrote the Gospels. The data may be the frequency of vocabulary or grammar used by the author of each section. In general, suppose that $X_{1}=\left(X_{11},\cdots ,X_{1m}\right),\cdots ,X_{K}=\left(X_{K1},\cdots ,X_{Km}\right)$ are *K* independent multinomial variables with parameter $\left(n_{1},p_{1}\right),\cdots ,\left(n_{K},p_{K}\right)$, where $p_{i}=\left(p_{i1},\cdots ,p_{im}\right)$, $\sum_{j=1}^{m}p_{ij}=1$ , i=1,⋯,K. On the ith point, there are $n_{i}$ experiments with *m* outcomes, and $X_{i}$ records the frequencies of *m* outcomes. We want to test
(1)$$H_{0}:p_{1}=\cdots =p_{K}\;v.s.\;H_{1}:p_{1}=\cdots =p_{k^{*}}\neq p_{k^{*}+1}=\cdots =p_{K}$$
where $k^{*}$ is the true change-point, $1<k^{*}<K$ . If $H_{0}$ is rejected, we further estimate $k^{*}$.

To solve this problem, Wolfe and Chen [16] proposed several statistics based on the cumulative sum (CUSUM) method. Horváth and Serbinowska [17] used the maximum likelihood ratio and maximum chi-square statistic to test the existence of change points and derived their transformed limit distribution. Batsidis and Horváth [18] extended it and proposed a family of phi-divergence tests that involves broad statistics. Riba and Ginebra [19] performed a graphical exploration of a sequence of polynomial observations and found a break point. Note that they all assumed that the number of categories *m* is fixed.

In recent years, the rise of big data has made the high-dimensional change-point problem more important. Thus, it becomes urgent to consider high-dimensional multinomial data as the categories of one thing in life can be quite large, such as the type of stores selling certain items on a shopping platform and the type of illness of patients in an outpatient clinic during a day. In this paper, we consider problem (Equation 1) with *m* tending to infinity. Recently, Wang et al. [20] proposed a procedure based on Pearson’s Chi-square test under the above scenario. Their idea is to pre-divide categories into two based on their probability magnitudes and to use the original and modified Pearson’s Chi-square statistic for large and small categories, respectively. This pre-classification can balance sparse and dense signals, resulting in good statistical performance. So, here, we use the pre-classification idea to construct a test statistic for problem (Equation 1) with *m* tending to infinity.

Another tool used in this article is based on information entropy. Entropy, originally a concept in statistical physics, was introduced into information theory by Shannon [21]. It has been widely applied in change-point problems. Unakafov and Keller [22] used ordinal mode conditional entropy to detect change points. Ma and Sofronov [23] proposed a cross-entropy algorithm to estimate the number and positions of change points. Vexler and Gurevic [24] applied empirical likelihood method to change-point detection, in which the essence of empirical likelihood estimation is a density-based entropy estimation. Mutual information, denoted by MI, computed as the difference between entropy and conditional entropy, is popular in deep learning, for example, see [25,26]. In the area of machine learning, MI is similar to information gain, which is often used as a measure of the goodness of a step in an algorithm, such as the selection of node splitting in a tree. Therefore, MI is naturally used as a metric in event detection problems. Relevant works include [27,28], etc. We utilize the MI between data and their position in this paper, given that a large value of MI means a high probability that a change point occurs.

In this paper, we consider the offline change-point problem. We propose a test statistic based on the mutual information for the at most one change-point (AMOC) problem (Equation 1) with *m* tending to infinity as the sample size tends to infinity. We adopt the pre-classification idea in [20] here. The optimal change-point position can also be estimated by MI. We show that the proposed statistic has an asymptotic normal distribution under the null distribution, and the power of the test converges to one under the alternative hypothesis. Meanwhile, we point out the relationship between MI and the likelihood ratio. In fact, the proposed statistic is based on the likelihood ratio method. As is widely acknowledged, although there is no uniformly most powerful test for change-point detection in general [29,30], the test based on the likelihood ratio structure has a high power [31]. Simulation studies demonstrate the excellent power of the test based on the proposed statistic as well as the high accuracy of the estimation. The innovations we have made in this article are that we replace the Pearson Chi-square statistic in Wang et al. [20] with mutual information and achieve better performance in terms of power and accuracy compared to their method.

The remaining structure of this paper is as follows. In Section 2, we present the proposed test statistic and the estimation method of a change point. In Section 3, we provide simulation results. In Section 4, we illustrate the method with an example based on physical examination data. In Section 5, we conclude the paper with some remarks. The proofs of the theorems are given in Appendix A.

## 2. Methods

### 2.1. Entropy and Mutual Information

We first briefly introduce some concepts about entropy and mutual information.

**Definition** **1.**
*Suppose that $x_{1},\cdots ,x_{u}$ are the possible values taken by a random variable X, where u can be infinity. Let $P_{X}\left(x_{i}\right)$ be the probability that $X=x_{i}$. The Shannon entropy of X is defined as*

(2)
$$H\left(X\right)=-\sum_{i=1}^{u}P_{X}\left(x_{i}\right)\log P_{X}\left(x_{i}\right).$$

*when $P_{X}\left(x_{i}\right)=0$, define $P_{X}\left(x_{i}\right)\log P_{X}\left(x_{i}\right)=0$.*


**Definition** **2.**
*Let Y be a random variable that takes values in $\left\{y_{1},\cdots ,y_{v}\right\}$, where v can be infinity. The conditional entropy of X given Y is defined as*

(3)
$$H\left(X|Y\right)=-\sum_{i=1}^{u}\sum_{j=1}^{v}P_{X,Y}\left(x_{i},y_{j}\right)\log P_{X|Y}\left(x_{i}|y_{j}\right),$$

*where $P_{X,Y}$, $P_{X|Y}$ are the joint probabilities of X and Y and the conditional probability of X given Y, respectively.*


**Definition** **3.**
*Assume that X and Y are the same as in Definitions 1 and 2. The mutual information (MI) of X relative to Y is defined as*

(4)
$$\bar{\mathrm{MI}}\left(X;Y\right)=\sum_{i=1}^{u}\sum_{j=1}^{v}P_{X,Y}\left(x_{i},y_{j}\right)\log\frac{P_{X,Y}\left(x_{i},y_{j}\right)}{P_{X}\left(x_{i}\right)P_{Y}\left(y_{j}\right)}.$$



The entropy value is larger when the data distribution is more symmetric. On the contrary, when the data are skewed, they have a small entropy [32]. The conditional entropy measures how much uncertainty is eliminated in *X* by observing *Y*. Obviously, mutual information can be written as the difference between entropy and conditional entropy, that is, $\bar{\mathrm{MI}}\left(X;Y\right)=H\left(X\right)-H\left(X|Y\right)$. It represents the average amount of information about X that can be gained or the amount of reduction of uncertainty in *X* by observing *Y*. $\bar{\mathrm{MI}}\left(X;Y\right)\geq 0$, and it becomes zero if *X* and *Y* are independent of each other.

### 2.2. Pre-Classification

For multinomial data, when the number of categories *m* is large, it is sometimes not realistic to treat all categories equally. For example, of all the cities in China, only a few of them account for half of the economy, which means that the rest of the cities have a small average share. The well-known Pareto principle [33] that 20% of the population owns 80% of the wealth in society also illustrates this phenomenon. Therefore, it is reasonable to classify the categories with different orders of magnitude.

Consider problem (Equation 1), i.e.,
$$H_{0}:p_{1}=\cdots =p_{K}\;v.s.\;H_{1}:p_{1}=\cdots =p_{k^{*}}\neq p_{k^{*}+1}=\cdots =p_{K}.$$
We denote $p_{1}=\cdots =p_{K}=q_{0}$ under $H_{0}$, and $p_{1}=\cdots =p_{k^{*}}=q_{0}$, $p_{k^{*}}=\cdots =p_{K}=q_{1}$ under $H_{1}$. Denote $q_{l}=\left(q_{l1},\ldots ,q_{lm}\right)$, l=0,1. Note that $q_{0}\neq q_{1}$.

Similar to Wang et al. [20], let $B_{0}$ be a subset of {1,…,m} such that $\max_{j\in B_{0}}q_{0j}a_{m}\to 0$, $B_{1}$ be a subset of {1,…,m} such that $\max_{j\in B_{1}}q_{1j}a_{m}\to 0$, where $a_{m}^{-1}$ is O(1) satisfying some conditions as m→∞. Let $A_{0}=B_{0}^{c}$ and $A_{1}=B_{1}^{c}$, where the superscript *c* stands for the complement operator. Assume that $\min_{j\in A_{0}}q_{0j}a_{m}>\varepsilon$ and $\min_{j\in A_{1}}q_{1j}a_{m}>\varepsilon$ for some ε>0 as m→∞. Let $A=A_{1}\cup A_{0}$ and $B=A^{c}$. Then, *m* categories are divided into large and small orders of magnitude by $a_{m}$ denoted by *A* and *B*. A change from $q_{0}$ to $q_{1}\neq q_{0}$ might occur either in *A* or *B*.

Let $X_{iA}$ be the component of $X_{i}$ in *A* for i=1,…,K and $q_{0A}$ be the component of $q_{0}$ in *A*. Let $X_{iB}$ and $q_{0B}$ be similarly defined. Then, the marginal distributions of $X_{iA}$ and $X_{iB}$ under the null assumption are
(5)$$\left(X_{iA},\sum_{j\in B}X_{ij}\right)∼\mathrm{Multi}\left(n_{i},\left(q_{0A},1-\sum_{j\in A}q_{0j}\right)\right),$$
and
(6)$$\left(X_{iB},\sum_{j\in A}X_{ij}\right)∼\mathrm{Multi}\left(n_{i},\left(q_{0B},1-\sum_{j\in B}q_{0j}\right)\right).$$

In the next subsection, we construct a statistic built on the marginal distributions (Equation 5) and (Equation 6).

Here are some additional notations. Denote $N=\sum_{i=1}^{K}n_{i}$, $N_{0k}=\sum_{i=1}^{k}n_{i}$, $N_{1k}=\sum_{i=k+1}^{K}n_{i}$ as the number of experiments in total, before and after time *k*, and $Z=\sum_{i=1}^{K}X_{i}$, $Z_{0k}=\sum_{i=1}^{k}X_{i}$, $Z_{1k}=\sum_{i=k+1}^{K}X_{i}$ as the number of successful trials in total, before and after time *k*. Let $\hat{q}=\frac{Z}{N}$, ${\hat{q}}_{0k}=\frac{Z_{0k}}{N_{0k}}$, and ${\hat{q}}_{1k}=\frac{Z_{1k}}{N_{1k}}$ be the corresponding frequencies.

For the data in *A*, let $Z_{A}=\sum_{i=1}^{K}X_{iA}$, $Z_{0kA}=\sum_{i=1}^{k}X_{iA}$, $Z_{1kA}=\sum_{i=k+1}^{K}X_{iA}$ be the number of successful trials in total, before and after time *k*. Define $Z_{BS}=\sum_{i=1}^{K}\sum_{j\in B}X_{ij}$, $Z_{0kBS}=\sum_{i=1}^{k}\sum_{j\in B}X_{ij}$, and $Z_{1kBS}=\sum_{i=k+1}^{K}\sum_{j\in B}X_{ij}$ as the sum of successful trials in *B* of total, before, and after *k*. Let ${\hat{q}}_{Aj}=\frac{Z_{Aj}}{N}$, ${\hat{q}}_{0kAj}=\frac{Z_{0kAj}}{N_{0k}}$, ${\hat{q}}_{1kAj}=\frac{Z_{1kAj}}{N_{1k}}$, j∈A, ${\hat{q}}_{BS}=\frac{Z_{BS}}{N}$, ${\hat{q}}_{0kBS}=\frac{Z_{0kBS}}{N_{0k}}$, ${\hat{q}}_{1kBS}=\frac{Z_{1kBS}}{N_{1k}}$ be the corresponding frequencies. Subscript *S* denotes the sum of frequencies. Similarly, we define $Z_{B}$, $Z_{0kB}$, $Z_{1kB}$, $Z_{AS}$, $Z_{0kAS}$, $Z_{1kAS}$, ${\hat{q}}_{Bj}$, ${\hat{q}}_{0kBj}$, ${\hat{q}}_{1kBj}$, j∈B, ${\hat{q}}_{AS}$, ${\hat{q}}_{0kAS}$, ${\hat{q}}_{1kAS}$. We illustrate some of the above notations in Table 1 in a more structured fashion.

### 2.3. Test Statistic

We use MI between the data $X=(X_{1},\ldots ,X_{K})$ and the location of the data to construct the statistic. For the data in *A*, the entropy is
(7)$$H_{A}=-{\hat{q}}_{BS}\log{\hat{q}}_{BS}-\sum_{j\in A}{\hat{q}}_{Aj}\log{\hat{q}}_{Aj}.$$

The entropies in *A* before and after *k* are
(8)$$H_{0kA}=-{\hat{q}}_{0kBS}\log{\hat{q}}_{0kBS}-\sum_{j\in A}{\hat{q}}_{0kAj}\log{\hat{q}}_{0kAj}$$
and
(9)$$H_{1kA}=-{\hat{q}}_{1kBS}\log{\hat{q}}_{1kBS}-\sum_{j\in A}{\hat{q}}_{1kAj}\log{\hat{q}}_{1kAj},$$
respectively.

Denote $Y^{k}=I\{$ the location of $X_{i}$ is before k} as the indicator function of the position of a sample relative to *k*. Note that, given the observations, $P\left(Y^{k}=1\right)=\frac{N_{0k}}{N}$ by the independence. By Section 2.1, the MI between *X* and $Y^{k}$ in *A* is
(10)$$H_{A}-H_{A}\left(X|Y^{k}\right)\hat{=}{\bar{\mathrm{MI}}}_{kA},$$
where $H_{A}\left(X|Y^{k}\right)=P\left(Y^{k}=0\right)H_{0kA}+P\left(Y^{k}=1\right)H_{1kA}=\frac{N_{0k}}{N}H_{0kA}+\frac{N_{1k}}{N}H_{1kA}$ is the conditional entropy of *X* given $Y^{k}$. Similarly, ${\bar{\mathrm{MI}}}_{kB}=H_{B}-\frac{N_{0k}}{N}H_{0kB}-\frac{N_{1k}}{N}H_{1kB}$, where $H_{B}$, $H_{0kB}$ and $H_{1kB}$ are defined similarly as in (Equation 7)–(Equation 9).

The uncertainty of *X* given $Y^{k}$ would reach the largest reduction if *k* is at the true break point $k^{*}$; hence, either ${\bar{\mathrm{MI}}}_{k^{*}A}$ or ${\bar{\mathrm{MI}}}_{k^{*}B}$ should be large. On the contrary, if the sequence is stable, the value of ${\bar{\mathrm{MI}}}_{k}$ should be small for any k∈{1,⋯,K}.

Since *A* and *B* are unknown, in light of Wang et al. [20], we use $\hat{A}=\left\{j:{\hat{q}}_{j}a_{m}>C\varepsilon \right\}$ to estimate *A*. Here, C>0 is some constant. As shown in [20], $\hat{A}$ is a consistent estimator of *A* if $a_{m}$ satisfies certain assumptions. Let $\hat{B}={\hat{A}}^{c}$. Construct the test statistic
(11)$$G_{m,\hat{A}}=\frac{2}{K}\sum_{k=1}^{K}\frac{N_{0k}N_{1k}}{N}{\bar{\mathrm{MI}}}_{k\hat{B}}+e_{m}I\left(\max_{k=1,\cdots ,K}2N{\bar{\mathrm{MI}}}_{k\hat{A}}>r_{m}\right)$$
for (Equation 1). Summation and maximization are conducted respectively for the MI of $\hat{A}$ and $\hat{B}$ in $G_{m,\hat{A}}$. The first term in $G_{m,\hat{A}}$ is the weighted log-likelihood ratio estimate, as pointed out after Lemma 1. The second term in $G_{m,\hat{A}}$ is based on the maximum norm of MI. It is widely acknowledged that the max-norm test is more suitable for sparse and strong signals, see [34,35]. $r_{m}$ is a threshold for $\hat{A}$, which ensures that the second term in $G_{m,\hat{A}}$ converges to zero under $H_{0}$. $e_{m}$ is a large number. Note that the statistic in [20] is based on the Pearson Chi-square statistic. Since in reality, the frequencies of small categories might be zeros, the Pearson Chi-square statistic for $\hat{B}$ is hence modified. The statistic presented here does not need to take into account the fact that a frequency may be zero, since by the definition of entropy, −plogp=0 if p=0. In order to study the properties of $G_{m,\hat{A}}$ better, we first give a lemma about MI .

**Lemma** **1.**
*Denote $L_{kA}=-2\log\left(\frac{\prod_{j\in A}{{\hat{q}}_{Aj}}^{Z_{Aj}}{{\hat{q}}_{BS}}^{Z_{BS}}}{{\prod_{j\in A}{\hat{q}}_{0kAj}}^{Z_{0kAj}}{{\hat{q}}_{0kBS}}^{Z_{0kBS}}{\prod_{j\in A}{\hat{q}}_{1kAj}}^{Z_{1kAj}}{{\hat{q}}_{1kBS}}^{Z_{1kBS}}}\right)$. Then, $2N{\bar{\mathrm{MI}}}_{kA}=L_{kA}$. It is also true by replacing A with B in all the subscripts, that is, $2N{\bar{\mathrm{MI}}}_{kB}=L_{kB}$.*


Note that $L_{kA}$ and $L_{kB}$ in Lemma 1 are estimations of minus two log likelihood ratios for data in *A* and *B* when the change-point is at *k*. Therefore, the problem based on MI can be transformed into the problem based on likelihood ratios.

By Lemma 1, the second term in (Equation 11) is $e_{m}I\left(\max_{k=1,\ldots ,K}L_{k\hat{A}}>r_{m}\right)$, and hence the existing limit theorems on likelihood ratios can be applied to it directly. The first term in (Equation 11) is $\frac{1}{K}\sum_{k=1}^{K}\frac{N_{0k}N_{1k}}{N^{2}}L_{k\hat{B}}$, which has the form of a weighted log likelihood ratio estimation. In Appendix A, we show that it is only an infinitesimal quantity away from some CUSUM statistic [36] using Taylor expansion and then prove the asymptotic distribution of $G_{m,\hat{A}}$ from related conclusions.

The sum of $L_{k\hat{B}}$ without weighting, $\sum_{k=1}^{K}L_{k\hat{B}}$, is closely related to the Shiryayev–Roberts procedure [37,38]. It uses $\sum_{k=1}^{K}\Lambda_{k}$ as a statistic, where $\Lambda_{k}$ is the likelihood ratio when the change point is at *k*. It is widely applied to determine the best stopping criterion in sequential change-point monitoring (see, e.g., [39]). However, replacing unknown parameters in $\sum_{k=1}^{K}\Lambda_{k}$ with their maximum likelihood estimation, which leads to $\sum_{k=1}^{K}L_{k\hat{B}}$ in this paper, would result in a complex asymptotic analysis [40]. So, here we use the weighted version $\frac{1}{K}\sum_{k=1}^{K}\frac{N_{0k}N_{1k}}{N^{2}}L_{k\hat{B}}$ instead of $\sum_{k=1}^{K}L_{k\hat{B}}$.

**Theorem** **1.**
*Let |A| denote the cardinality of any set A and sup|A| denote the maximal cardinality of the set A. Assume that sup|A|=d<∞, and*
*(i)* 
*$N{a_{m}}^{-2}{\left(\log a_{m}\right)}^{-1}\to \infty$ as (m,K)→∞,*
*(ii)* 
*$\frac{{\left[a(\log N)\right]}^{2}r_{m}}{{\left[b_{d}\left(\log N\right)\right]}^{2}}\to \infty$ and $\frac{\log e_{m}}{a\left(\log N\right)r_{m}^{1/2}-b_{d}\left(\log N\right)}\to 0$ as (m,K)→∞,*
*(iii)* 
*limsupK→∞m→∞max1≤k≤K2(nk+1N0k)12logN0k<∞,*

*limsupK→∞m→∞maxK2≤k≤K(nk+1N1k)12logN1k<∞.*

*Then, under $H_{0}$,*

$$\frac{G_{m,\hat{A}}-\frac{m-d+1}{6}}{\sqrt{\frac{m-d+1}{45}}}\overset{d}{⟶}N\left(0,1\right)$$

*as (m,K)→∞, where $a\left(x\right)={\left(2\log x\right)}^{\frac{1}{2}}$, $b_{d}\left(x\right)=2\log x+\left(d/2\right)\log\log x-\log\Gamma \left(d/2\right)$.*


Theorem 1 shows that $G_{m,\hat{A}}$ is asymptotically normally distributed under the null hypothesis. The condition (*i*) in Theorem 1 ensures the consistency of $\hat{A}$, which was also assumed in Theorem 1 of [20]. The condition (ii) in Theorem 1 requires the threshold $r_{m}$ to be large enough in order to guarantee that $e_{m}I\left(\max_{k=1,\ldots ,K}2N{\bar{\mathrm{MI}}}_{k\hat{A}}>r_{m}\right)$ converges to zero with probability one under the null hypothesis. Condition (iii) means that every $n_{i}$ is much less than *N*. Next, we focus on the properties of the statistic under the alternative hypothesis.

**Theorem** **2.**
*Assume that the conditions (i)–(iii) in Theorem 1 hold. Let $\delta_{j}=q_{1j}-q_{0j}$, j=1,⋯,m. Further assume that*
*(i)* 
*$e_{m}>cm$ as (m,K)→∞, where $c>\frac{1}{6}$.*
*(ii)* 
*$\frac{N_{k^{*}}}{N}\to \kappa_{0}$ as (m,K)→∞, $\kappa_{0}\in \left(0,1\right)$, and there exist $0<c_{1},c_{2}<\infty$ such that $c_{1}<\frac{N_{0k}}{N},\frac{N_{0k}}{N}<c_{2}$ for k=1,⋯,K as (m,K)→∞.*

*If the shift sizes $\delta_{j^{‘}}s$ satisfy either of the following two conditions,*
*(iii)* 
*$N{\delta_{j^{‘}}}^{2}{r_{m}}^{-1}\to \infty$ for some $j^{‘}\in A$,*
*(iv)* 
*$|\delta_{j^{‘}}|>0$ for some $j^{‘}\in B$,*

*then as (m,K)→∞,*

$$P\left(\left|\frac{G_{m,\hat{A}}-\frac{m-d+1}{6}}{\sqrt{\frac{m-d+1}{45}}}\right|>z_{1-\alpha }\right)\to 1,$$

*where $z_{1-\alpha }$ is the critical value of the standard normal distribution at level α.*


Theorem 2 establishes the consistency of the test under certain conditions when the probability in *A* or *B* changes. Condition (*i*) in Theorem 2 means that $e_{m}$ tends to infinity at a certain rate. It aims to ensure that $G_{m,\hat{A}}$ tends to infinity when the parameters in *A* change. Condition (ii) requires comparable sample sizes before and after the change point. The proofs of Theorem 1 and Theorem 2 are provided in Appendix A.

Once $H_{0}$ is rejected, we further use MI to estimate $k^{*}$. If $\max_{k=1,\ldots ,K}2N{\bar{\mathrm{MI}}}_{k\hat{A}}>r_{m}$, then $\hat{k}=\underset{k=1,\ldots ,K}{\arg\max}{\bar{\mathrm{MI}}}_{k\hat{A}}$; otherwise, $\hat{k}=\underset{k=1,\ldots ,K}{\arg\max}{\bar{\mathrm{MI}}}_{k\hat{B}}$. Numeric studies in the next section show that the power of the new statistic increases rapidly as the difference between the alternative hypothesis and the null hypothesis increases. At the same time, the precision of $\hat{k}$ using pre-classification is also satisfactory.

## 3. Simulation

We conduct simulation experiments to assess the performance of the test procedures in empirical size, power and estimation in finite samples. All results are based on 1000 replications. We use R to obtain simulation results. The necessary R code is given in Appendix B.

To analyze the empirical size, we simulate multinomial data with parameter $q_{0}=\left(\frac{\omega }{d}1_{d}^{⊤},\frac{1-\omega }{m-d}1_{m-d}^{⊤}\right)$ under the null hypothesis without break with reference to [20]. The first *d* probabilities are much greater than those of the latter. Hence, in reality, $\hat{A}$ can be chosen as $\left\{1,\ldots \ldots ,\hat{d}\right\}$. Following [20], we use $\hat{d}=\underset{i=1,\ldots ,m-1}{\arg\max}\frac{-1-{\hat{q}}_{\left(i\right)}{\hat{q}}_{\left(i+1\right)}}{\sqrt{1+{{\hat{q}}^{2}}_{\left(i\right)}}\sqrt{1+{{\hat{q}}^{2}}_{\left(i+1\right)}}}$, where ${\hat{q}}_{\left(1\right)}\geq \ldots \ldots \geq {\hat{q}}_{\left(m\right)}$ are the sorted values of ${\hat{q}}_{j}$. We consider different situations with the sample size *K* arranged from 50 to 500, and let m=K in each situation. For simplicity, we fix $n_{i}=100$, i=1,⋯,K. For the formula of $G_{m,\hat{A}}$, we choose $e_{m}=m$, $r_{m}={\left(2\log\log N+\frac{d}{2}\log\log\log N\right)}^{2}$ according to the conditions in the above section. The simulation results with various combinations of (ω, *d*) are reported in Table 2. We observe that the empirical size of the test is 4.5–6.7%, which is thus around the nominal 5% level in different situations. Here, we show the case of ω≤0.5. We also performed simulations for ω>0.5 and found empirical values slightly higher than 5% (data not shown).

To evaluate the power of the test, the alternative hypotheses stipulate a single break in the data sequence. We first consider parameters of two forms:(i)$q_{1}=\left\{\begin{array}{cc} \left(\left(1+s\right)\frac{\omega }{d}1_{\frac{d}{2}}^{⊤},\left(1-s\right)\frac{\omega }{d}1_{\frac{d}{2}}^{⊤},\frac{1-\omega }{m-d}1_{m-d}^{⊤}\right) & \;d\%2=0,\;\\ \left(\left(1+s\right)\frac{\omega }{d}1_{\frac{d-1}{2}}^{⊤},\frac{\omega [1-\frac{1+s}{2}\frac{d-1}{d}]}{\frac{d+1}{2}}1_{\frac{d+1}{2}}^{⊤},\frac{1-\omega }{m-d}1_{m-d}^{⊤}\right) & \;\mathrm{otherwise}; \end{array}\right.$(ii)$q_{1}=\left\{\begin{array}{cc} \left(\frac{\omega }{d}1_{d}^{⊤},\left(1+s\right)\frac{1-\omega }{m-d}1_{\frac{m-d}{2}}^{⊤},\left(1-s\right)\frac{1-\omega }{m-d}1_{\frac{m-d}{2}}^{⊤}\right) & \;(m-d)\%2=0,\;\\ \left(\frac{\omega }{d}1_{d}^{⊤},\left(1+s\right)\frac{1-\omega }{m-d}1_{\frac{m-d-1}{2}}^{⊤},\frac{\left(1-\omega \right)\left[1-\frac{1+s}{2}\frac{m-d-1}{m-d}\right]}{\frac{m-d+1}{2}}1_{\frac{m-d+1}{2}}^{⊤}\right) & \;\mathrm{otherwise}.\; \end{array}\right.$
ω is the proportion in *A*, 0<ω<1. $1_{d}$ denotes a *d*-dimensional vector with all components equal to 1. *s* represents the shift size. We consider different values of *s* when evaluating power and accuracy to better observe changes in efficiency and accuracy as the gap between the alternative and null hypotheses increases. % is the mod operation. The two alternative hypotheses assume that the change point is located in *A* and *B*, respectively. We consider $k^{*}=0.2K$ and 0.5K to capture breaks in the beginning and middle of a sample. For comparison, we use two competitors:The weighted maximum likelihood ratio statistic $L=\max_{k=1,\ldots ,K}\frac{N_{0k}N_{1k}}{N^{2}}{\hat{\Lambda }}_{k}$ where ${\hat{\Lambda }}_{k}=-2\log\left(\frac{\prod_{j=1,\ldots m}{{\hat{q}}_{j}}^{Z_{j}}}{{\prod_{j=1,\ldots m}{\hat{q}}_{0kj}}^{Z_{0kj}}{\prod_{j=1,\ldots m}{\hat{q}}_{1kj}}^{Z_{1kj}}}\right)$ proposed by Horváth and Serbinowska [17];The statistic $Q=\sum_{k=1}^{K}\sum_{j\in \hat{B}}\left(L_{kj}-{L_{kj}}^{\left(0\right)}\right)+e_{m}I\left(\max_{k=1,\ldots ,K}\max_{j\in \hat{A}}R_{kj}>r_{m}\right)$ in [20], in which $L_{kj}=\frac{N_{0k}N_{1k}}{N}{\left(\frac{Z_{0kj}}{N_{0k}}-\frac{Z_{1kj}}{N_{1k}}\right)}^{2}$ , ${L_{kj}}^{\left(0\right)}=\frac{N_{0k}N_{1k}}{N}\left(\frac{Z_{0kj}}{{N_{0k}}^{2}}+\frac{Z_{1kj}}{{N_{1k}}^{2}}\right)$ and $R_{kj}=\frac{L_{kj}}{{\hat{q}}_{j}}$, $e_{m}=K^{\frac{4}{3}}$ and $r_{m}=\log K\log K$ in the simulations.

The results are summarized in Figure 1 for level α=0.05. The size of *L* is on the high side, as seen from the curve at small *s* in Figure 1. The new test is very powerful, as evidenced by the rapid rate of convergence to 1 when *s* increases. In most cases, the empirical power of $G_{m,\hat{A}}$ is larger than the other two for alternative hypothesis (i). For the alternative hypothesis (ii), the three statistics perform equally well. These results further show that our test has higher power to detect a change located in the middle of the sample than in the beginning while the power is also still high.

We also briefly investigate how well the change-point location $k^{*}$ is approximated by the estimator $\hat{k}$ . We choose $k^{*}=0.2K$ and 0.5K as the change-point location. In Table 3 and Table 4, we report the mean and standard deviation of the absolute errors $|\hat{k}-k^{*}|$ for the different choices of *s* and *m* under the alternative hypothesis (i) or (ii), respectively. We compare our estimate with the maximum likelihood ratio estimate ${\hat{k}}_{L}=\underset{k=1,\ldots ,K}{\arg\max}{\hat{\Lambda }}_{k}$ and ${\hat{k}}_{Q}$ in [20].

The corresponding absolute errors in Table 3 and Table 4 underscore the considerable precision of $\hat{k}$, which improves when *s* is increased 0.3 from to 0.8. For the alternative (i), in almost all situations, $\hat{k}$ is better than the other two competitors. Small changes (for example, *s* = 0.3) are found with greater difficulty by using ${\hat{k}}_{L}$ and ${\hat{k}}_{Q}$, while the precision of $\hat{k}$ remains high. For the alternative hypothesis (ii), $\hat{k}$ and ${\hat{k}}_{L}$ have similar performance, and they are both slightly better than ${\hat{k}}_{Q}$. Alternative Hypothesis (i): Assume that the large probability changes while the small probability remains the same. Alternative Hypothesis (ii): Assume that the small probability changes while the large probability remains the same. Under alternative hypothesis (i), our method has better performance than the other two methods, probably because entropy as a non-linear function can increase the difference between frequencies, and it is more pronounced when the difference is small (e.g., *s* = 0.3).

Finally, we simulate the power and estimation precision for alternative hypothesis (iii): $$q_{1}=\left\{\begin{array}{cc} (\left(1+s\right)\frac{\omega }{d}1_{\frac{d}{2}}^{⊤},\left(1-s\right)\frac{\omega }{d}1_{\frac{d}{2}}^{⊤},\left(1+s\right)\frac{1-\omega }{m-d}1_{\frac{m-d}{2}}^{⊤},\left(1-s\right)\frac{1-\omega }{m-d}1_{\frac{p-d}{2}}^{⊤}), & d\%2=0,(m-d)\%2=0,\\ (\left(1+s\right)\frac{\omega }{d}1_{\frac{d-1}{2}}^{⊤},\frac{\omega [1-\frac{1+s}{2}\frac{d-1}{d}]}{\frac{d+1}{2}}1_{\frac{d+1}{2}}^{⊤},\left(1+s\right)\frac{1-\omega }{p-d}1_{\frac{m-d-1}{2}}^{⊤},\frac{\left(1-\omega \right)\left[1-\frac{1+s}{2}\frac{m-d-1}{m-d}\right]}{\frac{m-d+1}{2}}1_{\frac{m-d+1}{2}}^{⊤}), & \;\mathrm{otherwise},\; \end{array}\right.$$
where parameters in *A* and *B* change simultaneously, which was not mentioned in [20]. We compare our statistic and $\hat{k}$ with *Q* and ${\hat{k}}_{Q}$ in this case. The results are displayed in Figure 2, Table 5 and Table 6, from which we see that the power of $G_{m,\hat{A}}$ is slightly better than that of Q, and the precision of $\hat{k}$ is obviously higher than that of ${\hat{k}}_{Q}$.

## 4. Example

In this section, we use a data set to address the applicability of our method. The data concern the medical examination results of people working in Hefei’s financial sector (including banks and insurance companies) from 27 September 2017 to 25 August 2021, which includes each person’s age, gender, the date of examination and the disease detected. From the perspective of health analysis and disease prevention, it is thus important to understand the diseases in terms of how often they are detected.

Our goal is to test whether the proportion of people who have been diagnosed with some diseases change over time. After removing gender-specific diseases, we finally choose 210 diseases. Because in some weeks there is no person to have the examination, we eliminate those weeks and finally keep 173 weeks. Let $X_{i}$ be a 210-dimension vector with each component indicating the frequency of a certain disease detected during the $i_{th}$ time period. Then, there are K=173 vectors $X_{1},\cdots ,X_{173}$ of dimension m=210 with N=16596 outcomes.

Figure 3 shows the numbers of the top 30 diseases detected. The weekly sample size $n_{i}^{\prime }s$ are provided in Figure 4. We find from Figure 3 that the numbers of the first six diseases, Fatty Liver (FL), Overweight (OW), Thyroid Nodule (TN), Pulmonary Nodule (PN), Hepatic Cyst (HC) and Thyroid Cyst (TC), were much higher than those of the other diseases. By calculation, their proportions were, respectively, 0.088, 0.086, 0.06, 0.06, 0.038 and 0.032, which accounted for 35.8% of all the detected diseases. Hence, we choose $\hat{d}=6$. The value of the statistic is $G_{m\hat{A}}=200.2577$, and hence the null hypothesis that there is no change in the proportions of diseases detected is rejected.

Because $\max_{k=1,\ldots ,K}2N{\bar{\mathrm{MI}}}_{k\hat{A}}>r_{m}$, we find that $\hat{k}=\underset{k=1,\ldots ,K}{\arg\max}{\bar{\mathrm{MI}}}_{k\hat{A}}=14$, corresponding to 27 December 2017. This suggests that the proportions of diseases detected vary before and after 2018. Table 7 displays the proportions of the first six diseases before and after 2018. The proportions of Overweight and Thyroid Nodule were the highest before 2018. However, after 2018, the proportion of Fatty Liver jumped to the highest, and the proportion of Pulmonary Nodule also increased significantly. A possible explanation is that some unexpected events lead to changes in people’s lifestyles, which lead to changes in the proportion of the population suffering from different diseases. For example, the start of the Sino–US trade war in early February 2018 led to a continuous decline in the price of China’s A-shares, which was the trigger for the change in the lifestyle of financial practitioners after 2018. The study into the proportions of people with different diseases in the financial sector can reveal which disease is on the rise in this sector, and hence proper recommendations can be made for disease prevention.

## 5. Conclusions

This paper develops a change-point test based on MI for multinomial data when the number of categories is comparable to the sample size. We show that under certain conditions, the proposed statistic is asymptotically normal under the null hypothesis and consistent under the alternative hypothesis. The simulation results suggest that the test based on the proposed statistic has a high power. The proposed inference procedures are used to analyze the change in proportions of diseases detected in physical examination data during a period.

## Figures and Tables

**Figure 1 entropy-25-00355-f001:**
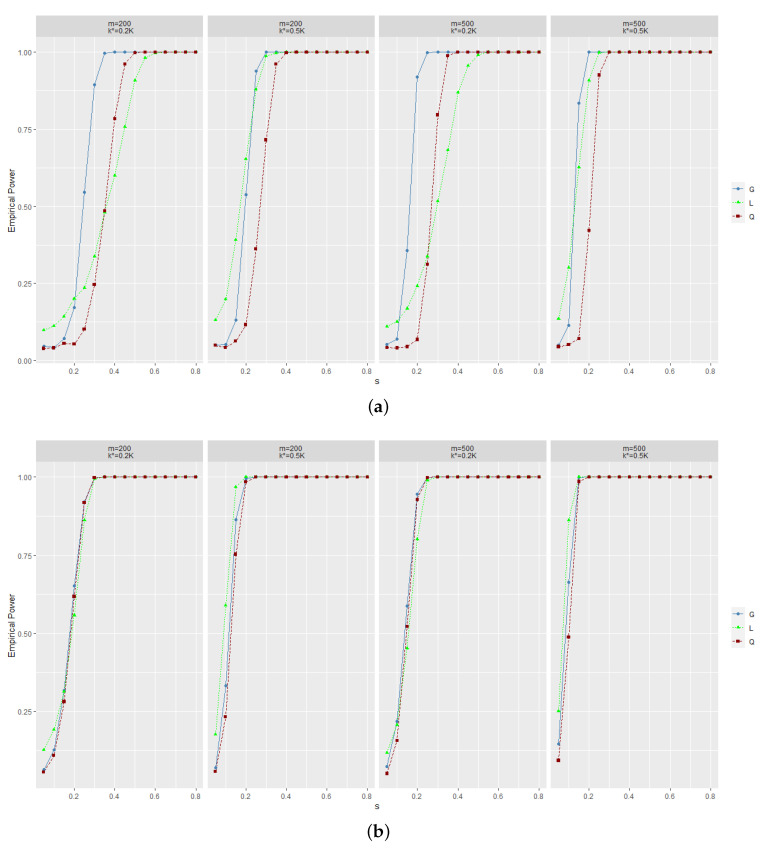
Empirical power of three statistics for level α=0.05. (**a**) The power under the alternative hypothesis (i). (**b**) The power under the alternative hypothesis (ii). *G* denotes the proposed statistic. *L* is the weighted maximum likelihood ratio statistic in [17]. *Q* is the statistic in [20]. ω=0.3, d=5.

**Figure 2 entropy-25-00355-f002:**
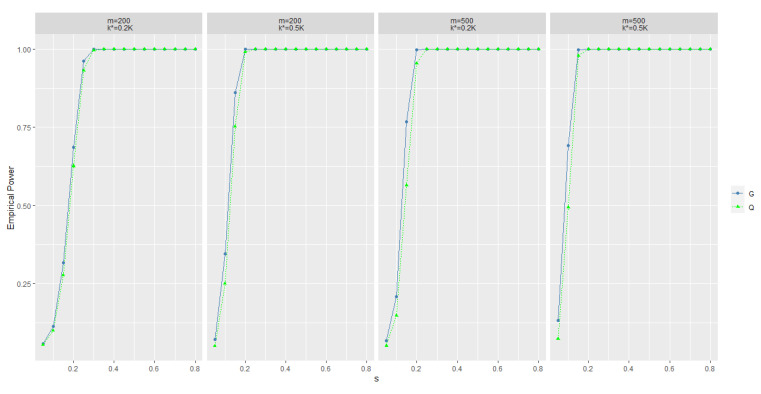
Empirical power of $G_{m\hat{A}}$ and *Q* under alternative hypothesis (iii). ω=0.3, d=5.

**Figure 3 entropy-25-00355-f003:**
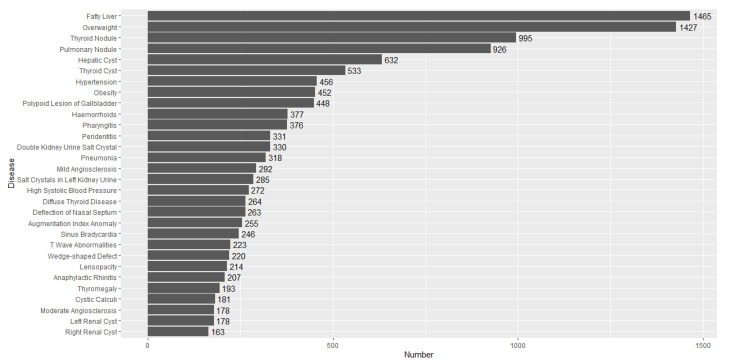
The numbers of top 30 diseases detected.

**Figure 4 entropy-25-00355-f004:**
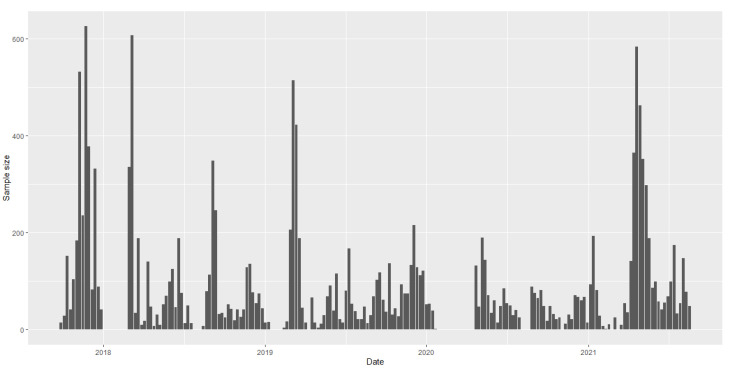
Weekly sample size.

**Table 1 entropy-25-00355-t001:** Explanation of some notations.

	Total		Before *k*		After *k*	
Experiment	*N*		$N_{0k}$		$N_{1k}$	
Category	*A*	*B*	*A*	*B*	*A*	*B*
Successful trials	$\left(Z_{A},Z_{BS}\right)$	$\left(Z_{B},Z_{AS}\right)$	$\left(Z_{0kA},Z_{0kBS}\right)$	$\left(Z_{0kB},Z_{0kAS}\right)$	$\left(Z_{1kA},Z_{1kBS}\right)$	$\left(Z_{1kB},Z_{1kAS}\right)$
Frequency	$\left({\hat{q}}_{A},{\hat{q}}_{BS}\right)$	$\left({\hat{q}}_{B},{\hat{q}}_{AS}\right)$	$\left({\hat{q}}_{0kA},{\hat{q}}_{0kBS}\right)$	$\left({\hat{q}}_{0kB},{\hat{q}}_{0kAS}\right)$	$\left({\hat{q}}_{1kA},{\hat{q}}_{1kBS}\right)$	$\left({\hat{q}}_{1kB},{\hat{q}}_{1kAS}\right)$

**Table 2 entropy-25-00355-t002:** Empirical sizes of $G_{m,\hat{A}}$ at the nominal test size 5% under different situations.

ω, *d*	*m*				
	50	100	200	300	500
(0.3, 5)	0.046	0.053	0.052	0.045	0.053
(0.3, 6)	0.052	0.058	0.059	0.052	0.052
(0.3, 10)	0.053	0.039	0.054	0.060	0.057
(0.5, 6)	0.050	0.034	0.047	0.066	0.067
(0.5, 8)	0.053	0.045	0.054	0.062	0.062
(0.5, 10)	0.053	0.051	0.061	0.066	0.054

**Table 3 entropy-25-00355-t003:** Mean and standard deviation (in parentheses) of $|\hat{k}-k^{*}|$, $|{\hat{k}}_{L}-k^{*}|$ and $|{\hat{k}}_{Q}-k^{*}|$ under the alternative hypothesis (i) or (ii) with ω=0.3, d=5, and $k^{*}=0.5K$.

*m*	*s*	Alternative Hypothesis (i)			Alternative Hypothesis (ii)		
		$|\hat{k}-k^{*}|$	$|{\hat{k}}_{L}-k^{*}|$	$|{\hat{k}}_{Q}-k^{*}|$	$|\hat{k}-k^{*}|$	$|{\hat{k}}_{L}-k^{*}|$	$|{\hat{k}}_{Q}-k^{*}|$
200	0.3	1.88(3.35)	15.95(27.15)	25.97(34.74)	0.64(1.16)	0.64(1.18)	0.66(1.20)
	0.4	0.77(1.32)	2.52(5.02)	3.94(6.90)	0.20(0.50)	0.20(0.50)	0.21(0.53)
	0.5	0.49(0.91)	1.11(1.93)	2.11(3.16)	0.06(0.25)	0.06(0.25)	0.06(0.27)
	0.6	0.31(0.67)	0.53(0.96)	1.67(2.59)	0.01(0.11)	0.01(0.11)	0.01(0.11)
	0.7	0.16(0.44)	0.29(0.63)	1.12(1.94)	0	0	0
	0.8	0.16(0.52)	0.16(0.48)	0.87(1.63)	0	0	0
500	0.3	1.57(2.32)	10.17(28.95)	6.40(9.60)	0.57(1.11)	0.56(1.10)	0.57(1.07)
	0.4	0.89(1.49)	2.24(3.41)	3.52(4.59)	0.16(0.46)	0.16(0.46)	0.18(0.48)
	0.5	0.50(0.92)	1.08(1.68)	2.18(3.15)	0.07(0.26)	0.07(0.26)	0.07(0.27)
	0.6	0.31(0.67)	0.51(1.06)	1.61(2.46)	0.01(0.11)	0.01(0.10)	0.02(0.13)
	0.7	0.15(0.40)	0.31(0.67)	1.10(1.83)	0	0	0
	0.8	0.09(0.32)	0.14(0.42)	0.88(1.39)	0	0	0

**Table 4 entropy-25-00355-t004:** Mean and standard deviation (in parentheses) of $|\hat{k}-k^{*}|$, $|{\hat{k}}_{L}-k^{*}|$ and $|{\hat{k}}_{Q}-k^{*}|$ under the alternative hypothesis (i) or (ii) with ω=0.3, d=5, and $k^{*}=0.2K$.

*m*	*s*	Alternative Hypothesis (i)			Alternative Hypothesis (ii)		
		$|\hat{k}-k^{*}|$	$|{\hat{k}}_{L}-k^{*}|$	$|{\hat{k}}_{Q}-k^{*}|$	$|\hat{k}-k^{*}|$	$|{\hat{k}}_{L}-k^{*}|$	$|{\hat{k}}_{Q}-k^{*}|$
200	0.3	10.01(31.28)	33.16(48.90)	66.82(58.79)	0.86(1.71)	0.87(1.70)	0.89(1.87)
	0.4	0.95(1.61)	8.21(23.85)	19.98(41.21)	0.24(0.59)	0.25(0.62)	0.25(0.60)
	0.5	0.73(0.26)	1.82(5.59)	2.45(3.94)	0.07(0.27)	0.08(0.28)	0.09(0.33)
	0.6	0.71(1.29)	0.75(1.35)	1.62(2.49)	0.02(0.15)	0.02(0.15)	0.03(0.18)
	0.7	0.45(0.88)	0.33(0.70)	1.2(2.17)	0	0	0
	0.8	0.30(0.66)	0.20(0.49)	0.93(1.73)	0	0	0
500	0.3	1.77(2.73)	52.01(105.27)	54.39(110.28)	0.80(1.29)	0.80(1.26)	0.83(1.46)
	0.4	0.94(1.48)	3.42(6.47)	3.82(5.64)	0.22(0.54)	0.23(0.56)	0.24(0.57)
	0.5	0.80(1.34)	1.40(2.27)	2.52(3.88)	0.07(0.28)	0.07(0.28)	0.07(0.28)
	0.6	0.68(1.22)	0.73(1.22)	1.56(2.27)	0.01(0.09)	0.01(0.09)	0.02(0.13)
	0.7	0.40(0.82)	0.36(0.75)	1.09(1.67)	0.01(0.08)	0.01(0.08)	0
	0.8	0.31(0.66)	0.20(0.48)	0.82(1.29)	0	0	0

**Table 5 entropy-25-00355-t005:** Mean and standard deviation (in parentheses) of $|\hat{k}-k^{*}|$ and $|{\hat{k}}_{Q}-k^{*}|$ under alternative hypothesis (iii) with ω=0.3, d=5 and $k^{*}=0.5K$.

*s*	m=200		m=500	
	$|\hat{k}-k^{*}|$	$|{\hat{k}}_{Q}-k^{*}|$	$|\hat{k}-k^{*}|$	$|{\hat{k}}_{Q}-k^{*}|$
0.3	1.82(2.63)	4.01(6.68)	1.62(2.68)	5.59(8.32)
0.4	0.82(1.42)	3.42(4.76)	0.82(0.82)	3.06(4.49)
0.5	0.47(0.89)	2.24(3.37)	0.46(1.07)	2.33 (3.33)
0.6	0.24(0.58)	1.52(2.48)	0.25(0.57)	1.37(2.15)
0.7	0.17(0.48)	1.10(1.72)	0.12(0.41)	1.16(1.77)
0.8	0.19(0.51)	0.83(1.31)	0.10(0.36)	0.84(1.42)

**Table 6 entropy-25-00355-t006:** Mean and standard deviation (in parentheses) of $|\hat{k}-k^{*}|$ and $|{\hat{k}}_{Q}-k^{*}|$ under alternative hypothesis (iii) with ω=0.3, d=5 and $k^{*}=0.2K$.

*s*	m=200		m=500	
	$|\hat{k}-k^{*}|$	$|{\hat{k}}_{Q}-k^{*}|$	$|\hat{k}-k^{*}|$	$|{\hat{k}}_{Q}-k^{*}|$
0.3	1.67(2.61)	1.70(3.69)	1.80(2.85)	4.89(7.62)
0.4	0.98(1.58)	3.22(6.08)	0.85(1.42)	3.59(5.02)
0.5	0.84(1.41)	2.44(3.78)	0.55(1.11)	2.54(3.80)
0.6	0.54(1.01)	1.54(2.41)	0.69(1.18)	1.67(2.45)
0.7	0.42(0.91)	1.28(2.12)	0.44(0.87)	1.16(1.91)
0.8	0.29(0.65)	0.88(1.40)	0.28(0.62)	0.87(1.47)

**Table 7 entropy-25-00355-t007:** Proportions of the first six diseases before and after $\hat{k}$.

	Disease	FL	OW	TN	PN	HC	TC
Proportion	before $\hat{k}$	0.078	0.095	0.100	0.021	0.035	0.045
	after $\hat{k}$	0.090	0.084	0.051	0.062	0.038	0.029

## Data Availability

Not applicable.

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
