# Peer review of "Change-Point Detection in a High-Dimensional Multinomial Sequence Based on Mutual Information"

_entropy, 2023, doi:10.3390/e25020355_

Round 1

Reviewer 1 Report

The authors proposed a statistic to test the existence of a change-point in a multinomial sequence. I list my comments as follows,

1. In Line 117, q_{im} should be q_{lm} in q_l. Place all the subscripts under operation max. Check the whole manuscript for this issue.

2. You applied the equation (11) to make a decision first, then use the idea in line 213-215 to find the change point. In fact, no matter the null hypothesis is reject or not, you still can find k* with this idea. How to explain/prove the statistic reject the null hypothesis is caused by k*?

3. In the simulation, you set up the q_0 with only two different values, ω/d or (1-ω)/(m-d), which is the very special situation. Why you did not consider the general case, such as, the first d components are (q_01,q_02,....q_0d), then the rest components have the same values?  Furthermore, you did not consider the case of  ω>0.5, why?

4. The two figures are too small to see their performances. You list m=50,100 in Table 1, however, you did not consider these two cases in Table 3 and 4. The position of the change point are always set in the middle of data, like 0.5K, how about the situations 0.2K or 0.8K?  In your application, k*=14, seems like 0.1K.

5. In the alternative hypothesis (iii), although Wang [16] did not consider it, you can use their method to get the corresponding results, which would be consistent with the table 2. Generally speaking, you need to explain more findings in details after these tables, for example, but not limited to, why you consider different values of s here? Why all the statistics have the similar/same performance under alternative hypothesis (ii)? Why the mean and standard deviation of k from different methods are larger under alternative hypothesis (i) for s=0.3?  

6. In your application, what are the hypothesis based on the real problem?  How to understand this change point in combination with the reality? 

7. What software do you use to obtain the results in simulation and application? You need to put the necessary code in Appendix. 

Reviewer 2 Report

In many real applications, observations are taken sequentially over time, or can be ordered with respect to some other criterion. The main question is whether the data obtained are generated using one or several different statistical models, and, therefore, there is a need to identify so-called change-points. The change-point problem can be considered one of the most interesting problems in statistics and it arises in a wide variety of fields, including bioinformatics, biomedical signal processing, speech and image processing, seismology, industry (e.g. fault detection) and financial mathematics. The manuscript proposes a new statistic to test the existence of a change-point in a multinomial sequence, where the number of categories is comparable with the sample size as it tends to infinity. The authors show that the proposed statistic is asymptotically normal under the null distribution and the power of the test converges to one under the alternative hypothesis. The performance of the proposed method is illustrated using both simulated and real data sets.

I have a few comments on the manuscript.

Abstract and Introduction. While you cite some papers on multiple change-point detection (for example, [3] and [14]), make sure that you stress the fact that in this paper you consider a single change-point detection problem. You also need to mention that you consider the posterior (or offline) change-point problem compare to the sequential (also, quickest or online) problem.

Line 14. Insert a space between “Page” and “[1]”.

Line 16. “and so on” may not look appropriate here. For example, you could extend the list of possible applications.

Lines 43-44. Probably, it should be “derived” instead of “proved” in “proved their transformed limit distribution”, unless you want to continue this sentence and mention the limiting distribution.

Line 45. Give more details instead of just saying “Other references include [14,15], etc”. Do not use “etc” in a situation like this. I would recommend that you could cite a few recent review papers or books on change-point detection.

Line 62. Replace “position” by “positions”.

Line 74 (see also lines 52, 156-157, 162, 194, for example). It appears that this manuscript heavily relies on the results of the paper by Wang et al. [16]. Could the authors clearly explain in the Introduction what innovations they have made compare to [16]?

Line 79. Introduce the abbreviation “UMP” here since not all readers may know that it means “uniformly most powerful”.

Line 86. Replace “concludes” by “conclude”.

Line 87. Insert “the” before “theorems”.

Line 91. Is “m” here the same as the “m” used in the Introduction (for example, see lines 37-40)? If not, you should use a different notation to avoid any confusion.

Formula (2). Is “log” the natural logarithm here?

Line 107. By “it vanishes” do you mean the MI becomes 0?

Lines 112-113. It would be good to have a reference to this principle here.

Line 117. What are “q_l1” and “q_im”? Shouldn’t it be “q_lm” instead of “q_im”?

Lines 118-120. It is not quite clear how “a_m” is defined.

Line 120. Are “B_0^c” and “B_1^c” the compliments of “B_0” and “B_1”?

Lines 130-140. To be honest, the notations introduced here are difficult to follow. Would it be possible to illustrate (at least some of) them on a plot or to put them in a table in a more structured way?

Formula (10). The notations are not clear here. What does “^” above “=” mean? Why is the line above “MI” needed here? If it is a mutual information, how it is related to Definition 3?

Line 157. Probably, it should be “estimator” instead of “estimate”.

Line 160. Explain what the max-norm test is.

Lines 160-162. How are the “r_m” and “e_m” specified (that is, what values of r_m and e_m should be used to calculate the value of the test statistic)?

Lines 176-177. Insert a reference to the CUSUM statistic here.

Lines 179-180. Insert references to the original papers by Shiryaev and Roberts here.

Line 186. Explain this statement “sup |A|=d”. What does it mean if “A” is a set? The maximal element in a set?

Theorem 1 (see also Theorem 2). What would be possible choices of “a_m”, “r_m” and “e_m” satisfying these conditions? It would be good to give some insights on how to select these sequences.

Line 222-223. It is not clear how parameter “q_0” is defined here. What are the omega and the bold 1? If the bold 1 is an identity matrix, why is it transposed?

Lines 228-229. Why this choice of “r_m” and “e_m”? It would be good to see how different choices of “r_m” and “e_m” affect the performance of the proposed method.

Line 233. Probably, it should be “power” instead of “powers”.

Figures 1, 2, 3 and 4. The figures should be made bigger since it is very difficult to read them (also think about increasing the font size).

Line 272. “sectors” -> “sector”.

Line 273. It is better to say “including banks and insurance companies”. Do not use “etc.” in situations like this.

Line 327. Indicate what lemma or theorem from [16] corresponds to Lemma A1. Probably, for consistency, it might be better to say “Proof of Lemma A1” instead of “Proof”.

Reviewer 3 Report

In general, the manuscript is interesting and could be recommended for publication in the Journal after a minor revision. 

The change-point detection is a classical problem in time series analysis. The authors provide a literature overview and present their approach based on mutual information. Some computational experiments are used to illustrate the presented technique. 

The authors are recommended to expand the literature overview. The change-point detection can be employed as a tool in a much more demanding task (time series segmentation). A typical reference in this area of research (when change-point is exploited for the determination of the end of a segment) is presented in [A]. A discussion on these issues would enrich the literature overview and highlight the importance of the topic. 

Also, the authors could extend the computational part by adding more comparisons with alternative techniques for the detection of the change-point. 

[A] Algebraic segmentation of short nonstationary time series based on evolutionary prediction algorithms. Neurocomputing (2013) vol.121, p.354-364.

Round 2

Reviewer 1 Report

Thank you for your reply. I think I can recommend this paper to be published in this journal.

By the way, please only keep the modified part and delete the original content that needs to be modified.

Reviewer 2 Report

The authors have addressed the comments made by the reviewers.